# Speech Synthesis By Unrolling Diffusion Process using Neural Network Layers

**Peter Ochieng**
*Department of Computer Science and Technology*
*University of Cambridge*

*po304.cam.ac.uk*

## Abstract

This work introduces UDPNet, a novel architecture designed to accelerate the reverse diffusion process in speech synthesis. Unlike traditional diffusion models that rely on timestep embeddings and shared network parameters, UDPNet unrolls the reverse diffusion process directly into the network architecture, with successive layers corresponding to equally spaced steps in the diffusion schedule. Each layer progressively refines the noisy input, culminating in a high-fidelity estimation of the original data, $x_0$. Additionally, we redefine the learning target by predicting latent variables instead of the conventional $x_0$ or noise $\epsilon_0$. This shift addresses the common issue of large prediction errors in early denoising stages, effectively reducing speech distortion. Extensive evaluations on single- and multi-speaker datasets demonstrate that UDPNet consistently outperforms state-of-the-art methods in both quality and efficiency, while generalizing effectively to unseen speech. These results position UDPNet as a robust solution for real-time speech synthesis applications. Sample audio is available at `https://onexpeters.github.io/UDPNet/`.

## 1 Introduction

Diffusion Probabilistic Models (DPMs) (Sohl-Dickstein et al., 2015) have become a dominant paradigm in speech synthesis (Lam et al., 2022; Chen et al., 2020; Kong et al., 2020b) due to their ability to model complex data distributions. These models operate through two key stages: (1) a **forward process**, where Gaussian noise is gradually added to the input data until it becomes indistinguishable from white noise, and (2) a **reverse process**, where a neural network iteratively removes noise to reconstruct the original signal.

While effective, **DPMs suffer from slow inference**, requiring hundreds to thousands of iterative reverse steps per sample. Various strategies, such as noise schedule optimization (Chen et al., 2020) and scheduling networks (Lam et al., 2022), have attempted to accelerate sampling. However, even these methods remain computationally prohibitive for real-time applications like text-to-speech (TTS) and voice cloning.

To address this, we propose **UDPNet**, a novel *layer-wise diffusion model* that restructures the denoising process within the network itself. Unlike traditional diffusion models, which use a **single shared model** across all timesteps, UDPNet **assigns successive layers to progressively refine the signal over coarser time intervals**. This structured refinement preserves the multi-step denoising behavior of diffusion models while significantly improving efficiency. Importantly, UDPNet **does not increase model parameters**, as it shares weights across layers rather than learning separate parameters for each timestep. The number of layers $N$ is determined by a skip parameter $\tau$:

$$N = \frac{T}{\tau}, \tag{1}$$

where $T$ is the total number of forward diffusion steps. Each layer incrementally refines the noisy representation, reducing the need for sequential sampling and lowering computational cost.

A fundamental limitation of standard diffusion models is that early timesteps contain highly degraded representations of $x_0$, making direct reconstruction difficult. The total expected reconstruction error in traditional diffusion can be expressed as:

$$\mathcal{E}_{\text{Standard}} = \sum_{t=1}^{T} \mathbb{E}\left[\|x_0 - x_\theta(x_{t+1}, t)\|^2\right].$$

Since $x_0$ contains the most structured information, directly regressing toward $x_0$ at early timesteps forces the model to learn **unrealistic mappings** from highly noisy inputs, leading to large prediction errors. In **full-step denoising**, this issue is mitigated as later iterations **gradually refine predictions**, while **clipping of** $z_t$ limits error propagation. However, in **few-step denoising**, early-stage errors persist longer due to **fewer refinement opportunities**. This cascading effect not only **complicates training** but also degrades the quality of generated speech, often introducing perceptual artifacts—particularly in speech synthesis (Zhou et al., 2023).

Thus, **minimizing the initial error gap is crucial** for maintaining **high-quality reconstruction** in reduced-step denoising. To address this, UDPNet introduces a layer-wise denoising strategy, ensuring that each layer predicts a **closer latent state** $x_t$ instead of directly regressing toward $x_0$. The expected accumulated error in our approach is given by:

$$\mathcal{E}_{\text{Layer-Wise}} = \sum_{l=1}^{N} \mathbb{E}\left[\|x_t - x_\theta^{(l)}(\hat{x}_{t+\tau})\|^2\right], \quad N = \frac{T}{\tau}. \tag{2}$$

This structured denoising approach reduces large per-step reconstruction gaps, stabilizes training, and enhances sample fidelity, making UDPNet well-suited for efficient few-step denoising. This formulation reduces the per-step reconstruction gap and stabilizes training. Specifically, UDPNet offers the following advantages:

- **Progressive noise removal:** Each layer gradually removes a fraction of the noise, avoiding large per-step reconstruction errors (see Figure 2).

- **Reduced prediction gap:** Since $x_t$ is closer to $x_\theta^{(l)}$ at each layer, the **accumulated error is lower**.

- **Smoother transitions:** Intermediate layers prevent the model from amplifying early-stage prediction errors.

- **Enhanced stability in early denoising stages:** Predicting $x_0$ from highly degraded inputs is error-prone. Instead, predicting intermediate latent variables allows the model to refine coarse structures before reconstructing finer details.

As a result, UDPNet achieves **sub-linear error accumulation**, leading to more stable denoising:

$$\mathcal{E}_{\text{Layer-Wise}} = O\left(\frac{T}{\tau}\right) \quad \text{vs.} \quad \mathcal{E}_{\text{Standard}} = O(T).$$

In summary, UDPNet adds the following contributions:

- Faster inference speeds compared to traditional diffusion models, making it feasible for real-time speech applications.

- Improved speech quality, as validated through subjective and objective evaluation metrics.

- Strong generalization across unseen speakers and datasets.

## 2 Background

### 2.1 Denoising Diffusion Probabilistic Model (DDPM)

Given an observed sample $x$ of unknown distribution, DDPM defines a forward process as:

$$q(x_{1:T}|x_0) = \prod_{i=1}^{T} q(x_t|x_{t-1}) \tag{3}$$

Here, latent variables and true data are represented as $x_t$ with $t = 0$ being the true data. The encoder $q(x_t|x_{t-1})$ seeks to convert the data distribution into a simple, tractable distribution after $T$ diffusion steps.

The encoder models the hidden variables $x_t$ as linear Gaussian models with mean and standard deviation centered around its previous hierarchical latent $x_{t-1}$. The mean and variance can be modeled as hyperparameters (Ho et al., 2020) or as learnable variables (Nichol & Dhariwal, 2021; Kingma et al., 2021). The Gaussian encoder's mean and variance are parameterized as:

$$u_t(x_t) = \sqrt{\alpha_t} x_{t-1}, \quad \Sigma_q(x_t) = (1 - \alpha_t)I \tag{4}$$

Thus, the encoder can be expressed as:

$$q(x_t|x_{t-1}) = \mathcal{N}(x_t; \sqrt{\alpha_t} x_{t-1}, (1 - \alpha_t)I) \tag{5}$$

where $\alpha_t$ evolves with time $t$ based on a fixed or learnable schedule such that the final distribution $p(x_T)$ is a standard Gaussian.

Using the property of isotropic Gaussians, (Ho et al., 2020) show that $x_t$ can be derived directly from $x_0$ as:

$$x_t = \sqrt{\bar{\alpha}_t} x_0 + \sqrt{(1 - \bar{\alpha}_t)} \epsilon_0 \tag{6}$$

where $\bar{\alpha}_t = \prod_{t=1}^{t} \alpha_t$ and $\epsilon_0 \sim \mathcal{N}(0, I)$. Hence,

$$q(x_t|x_0) = \mathcal{N}(x_t; \sqrt{\bar{\alpha}_t} x_0, (1 - \bar{\alpha}_t)I). \tag{7}$$

The reverse process, which seeks to recover the data distribution from the white noise $p(x_T)$, is modeled as:

$$p_\theta(x_{0:T}) = p(x_T) \prod_{i=1}^{T} p_\theta(x_{t-1}|x_t) \tag{8}$$

where $p(x_T) = \mathcal{N}(x_T; 0, I)$. The goal of DPM is to model the reverse process $p_\theta(x_{t-1}|x_t)$ so that it can be used to generate new data samples.

### 2.2 Optimization of DPM

After the DPM has been optimized, a sampling procedure entails: 1. Sampling Gaussian noise from $p(x_T)$. 2. Iteratively running the denoising transitions $p_\theta(x_{t-1}|x_t)$ for $T$ steps to generate $x_0$.

To optimize DPM, the evidence lower bound (ELBO) is used:

$$\log p(x) = E_{q(x_0)}[D_{KL}(q(x_T|x_0)||p(x_T))+$$
$$\sum_{t=2}^{T} E_{q(x_t|x_0)}[D_{KL}(q(x_{t-1}|x_t,x_0)||p_\theta(x_{t-1}|x_t))]- \tag{9}$$
$$E_{q(x_1|x_0)}[\log p_\theta(x_0|x_1)].$$

In Eq. 9, the second term on the right is the denoising term that seeks to model $p_\theta(x_{t-1}|x_t)$ to match the ground truth $q(x_{t-1}|x_t,x_0)$. In (Ho et al., 2020), $q(x_{t-1}|x_t,x_0)$ is derived as:

$$q(x_{t-1}|x_t,x_0) = \mathcal{N}\left( \frac{\sqrt{\alpha}(1-\bar{\alpha}_{t-1})x_t + \sqrt{\bar{\alpha}_{t-1}}(1-\alpha_t)x_0}{(1-\bar{\alpha}_t)}, \right.$$
$$\left. \frac{(1-\alpha_t)(1-\bar{\alpha}_{t-1})}{(1-\bar{\alpha}_t)}I \right). \tag{10}$$

To ensure $p_\theta(x_{t-1}|x_t)$ matches $q(x_{t-1}|x_t,x_0)$, it is modeled with the same variance $\Sigma_q(t)$, given by:

$$\Sigma_q(t) = \frac{(1-\alpha_t)(1-\bar{\alpha}_{t-1})}{(1-\bar{\alpha}_t)}I. \tag{11}$$

The mean of $p_\theta(x_{t-1}|x_t)$ is parameterized as:

$$u_\theta(x_t,t) = \frac{\sqrt{\alpha}(1-\bar{\alpha}_{t-1})x_t + \sqrt{\bar{\alpha}_{t-1}}(1-\alpha_t)\hat{x}_\theta(x_t,t)}{(1-\bar{\alpha}_t)}. \tag{12}$$

Here, the score network $\hat{x}_\theta(x_t,t)$ is parameterized by a neural network that seeks to predict $x_0$ from a noisy input $x_t$ and time index $t$. Thus,

$$p_\theta(x_{t-1}|x_t) = \mathcal{N}(\mu_\theta, \Sigma_q(t)). \tag{13}$$

Optimizing the KL divergence between $q(x_{t-1}|x_t,x_0)$ and $p_\theta(x_{t-1}|x_t)$ can be formulated as:

$$L_{t-1} = \arg\min_\theta E_{t\sim U(2,T)} D_{KL}(q(x_{t-1}|x_t,x_0)||p_\theta(x_{t-1}|x_t)). \tag{14}$$

Using standard derivations (Luo, 2022), we obtain:

$$L_{t-1} = \arg\min_\theta E_{t\sim U(2,T)}||\hat{x}_\theta(x_t,t) - x_0||_2^2. \tag{15}$$

By rearranging Eq. 6, we obtain:

$$x_0 = \frac{x_t - \sqrt{1-\bar{\alpha}_t}\epsilon_0}{\sqrt{\bar{\alpha}_t}}. \tag{16}$$

Thus, an equivalent optimization using a noise predictor $\hat{\epsilon}_\theta(x_t,t)$ is:

$$L_{t-1} = \arg\min_\theta E_{t\sim U(2,T)}||\hat{\epsilon}_\theta(x_t,t) - \epsilon_0||_2^2. \tag{17}$$

Work in (Ho et al., 2020) shows that $L_{t-1}$ optimizes the ELBO.

# 3 Related Work

## 3.1 Acceleration of Diffusion Models

Recent efforts to accelerate diffusion models have focused on two primary strategies: **distribution matching methods** and **consistency models**.

### 3.1.1 Distribution Matching and Knowledge Distillation Approaches

Several works (Yin et al., 2024; Xiao et al., 2021; Sauer et al., 2024; Luo et al., 2024; Salimans & Ho, 2022) aim to accelerate sampling by distilling the knowledge of a **pre-trained diffusion model** into a lightweight student model. These methods train student models to match the marginal distributions of a pre-trained teacher model. While effective, they introduce several challenges:

- **Dependence on Pre-Trained Models:** These methods require a computationally expensive teacher model, limiting their adaptability to new datasets.

- **Distillation Sensitivity:** The effectiveness of knowledge distillation heavily depends on hyperparameter tuning, making training computationally demanding.

- **Mode Collapse Risk:** Single-step denoising models may suffer from reduced sample diversity, as they risk collapsing into a limited subset of the data distribution.

### 3.1.2 Consistency Models and Latent Space Acceleration

More recent approaches, such as Consistency models (Song et al., 2023; Li et al., 2024) and latent consistency models (LCMs) (Luo et al., 2023), aim to improve efficiency by learning a direct mapping from noise to clean data. These models enforce consistency constraints across diffusion steps, allowing for near single-step sampling without iterative refinement. Despite their promising acceleration benefits, consistency models exhibit notable trade-offs:

- **Constraint-driven regularization:** Consistency models require strict regularization to ensure smooth interpolation, which can degrade sample quality in complex domains such as speech synthesis.

- **Loss of multi-step refinement:** By enforcing single-step consistency, these models may forgo the iterative refinements that contribute to high-quality generative performance.

- **Performance in speech synthesis:** While consistency models have been explored in image synthesis, their adaptation to high-dimensional, temporally correlated data such as speech remains an open challenge.

### 3.1.3 Scheduling-Based Optimizations

An alternative acceleration strategy is **noise schedule optimization**, as explored in WaveGrad (Chen et al., 2020) and BDDM (Lam et al., 2022), where adaptive schedules reduce the number of required sampling steps. However, despite these advancements, **diffusion models remain computationally expensive, particularly for real-time applications such as speech synthesis**.

### 3.1.4 Our Approach: UDPNet for Efficient Diffusion Acceleration

Unlike previous methods that rely on distillation, consistency constraints, or noise schedule optimizations, our approach **eliminates the need for a pre-trained model** and instead optimizes inference-time efficiency directly within the training framework. UDPNet restructures the denoising process within the network's architecture itself, reducing iterative computations while maintaining multi-step refinement.

Key advantages of **UDPNet** over existing methods:

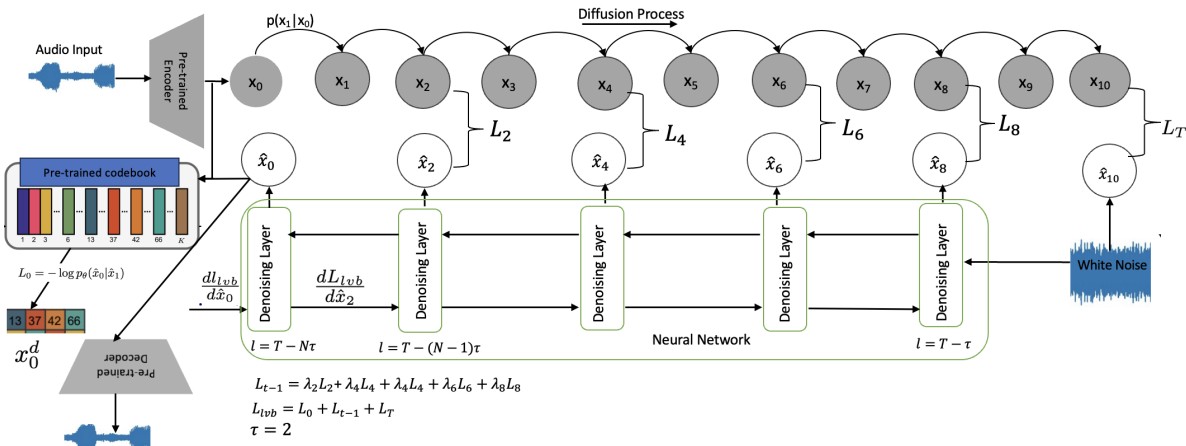

Figure 1: An overview of the unconditioned audio generation. An input audio is processed by a pre-trained model to generate $x_0$. $x_0$ is then processed by forward process to generate latent variables $x_t$. In the reverse process, white noise $x_T$ is passed through the first layer of the neural network and processed through the subsequent layers. A layer is mapped to a given time step $t$ of the forward process. If a layer $l$ is mapped to a time step $t$, an error $L_i$ is computed by establishing $l_2$ norm between their respective embeddings.

- **No reliance on pre-trained models** – Unlike distillation-based techniques, our approach does not require an external teacher model, making it more adaptable to new datasets.

- **Maintains progressive refinement** – Unlike consistency models, UDPNet preserves multi-step refinement while accelerating sampling.

- **Optimized for real-time speech synthesis** – Unlike previous methods primarily designed for images, our approach is tailored for efficient speech denoising.

## 4  Speech synthesis by Unrolling diffusion process using Neural network layers

### 4.1  Unconditional Speech Generation

#### 4.1.1  Forward Process

During the forward process, a raw audio waveform $x$ is encoded by a pre-trained encoder into its latent representation $x_0 \in \mathbb{R}^{f \times h}$, where $f$ denotes the number of frames and $h$ is the hidden dimension size. This representation serves as the foundation for generating latent variables $x_t$ through the forward diffusion process, as described in Equation 6, for $1 \le t \le T$.

To facilitate the reconstruction process during the reverse diffusion, a pre-trained discrete codebook of size $K$ with dimension $h$ is employed. The codebook, denoted as $\mathcal{Z} = \{z_k\}_{k=1}^{K} \in \mathbb{R}^h$, maps each row of $x_0$ to the closest entry in the codebook. This mapping is determined by minimizing the squared Euclidean distance, as shown in Equation 18:

$$z_q = \left( \underset{z_k \in \mathcal{Z}}{\arg\min} \, \|x_0^i - z_k\|_2^2 \; \forall i \in f \right) \in \mathbb{R}^{f \times h}. \tag{18}$$

Here, $z_q$ represents the quantized latent representation, where each row $x_0^i$ is replaced by the nearest codebook vector $z_k$. These discrete indices, $x_0^d$, correspond to the assigned codebook entries for each row of $x_0$. The stored indices play a critical role in the reverse diffusion process, enabling the recovery of the original signal from the latent representation(we discuss recovery in the next section).

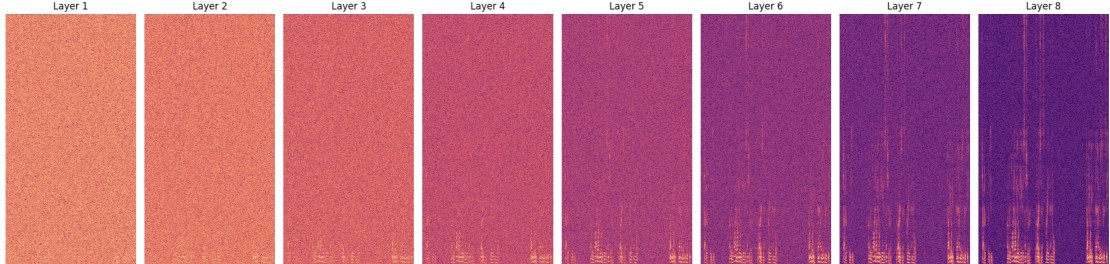

(i): Denoising progression in an 8-layer model. Noise is gradually removed in small steps, resulting in a smoother and more natural reconstruction of the speech signal.

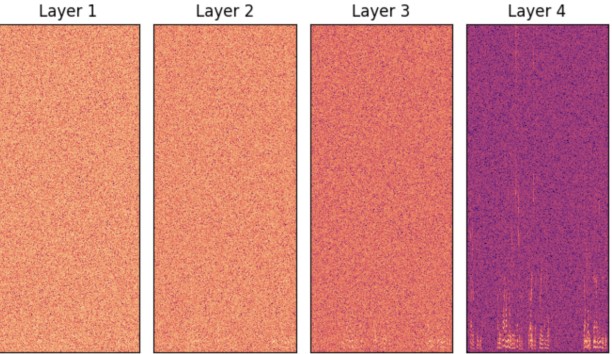

(ii): Denoising progression in a 4-layer model. The noise reduction is less gradual, leading to more abrupt changes and artifacts in the final output.

Figure 2: Comparison of denoising performance between an 8-layer model (i) and 4-layer model (ii). The 8-layer model progressively refines intermediate representations $x_t$, leading to a more stable and natural reconstruction. In contrast, the 4-layer model exhibits more abrupt noise reduction, highlighting the importance of a gradual denoising process where the model refines $x_t$ rather than directly predicting $x_0$.

### 4.1.2 Reverse Process

The ELBO (Equation 9) used for optimizing diffusion probabilistic models consists of three key parts:

$$L_0 = \mathbb{E}_{q(x_1|x_0)} \left[ \log p_\theta(x_0|x_1) \right]$$
$$L_T = \mathbb{E}_{q(x_0)} \, \mathrm{D_{KL}} \left( q(x_T|x_0) \| p(x_T) \right)$$
$$L_{t-1} = \sum_{t=2}^{T} \mathbb{E}_{q(x_t|x_0)} \left[ \mathrm{D_{KL}} \left( q(x_{t-1}|x_t, x_0) \| p_\theta(x_{t-1}|x_t) \right) \right]$$

The total loss, based on these three parts, is defined as:

$$L_{vlbfull} = L_0 + \sum_{t=1}^{T-1} L_{t-1} + L_T \tag{19}$$

The term $L_{t-1}$, also known as the denoising term, is critical for teaching the model to estimate the transition $p_\theta(x_{t-1}|x_t)$, which approximates the true distribution $q(x_{t-1}|x_t, x_0)$. Minimizing the KL divergence between

these two distributions ensures that the model can effectively remove noise and progressively recover the original data.

In traditional diffusion models, each timestep of the reverse process requires passing noisy data through a neural network multiple times, making inference computationally expensive. Our model replaces this by distributing the denoising process across the network layers, where each layer is responsible for refining the signal over a specific range of timesteps. This significantly reduces computational overhead while preserving sample quality.

To achieve this, we introduce **deterministic approximations** $\hat{x}_{t-1}$ and $\hat{x}_t$ in place of stochastic latent variables. Unlike traditional diffusion models that explicitly **sample Gaussian noise** at each reverse step, our approach **implicitly models noise transitions** through the hierarchical structure of the neural network. This eliminates redundant noise sampling while **preserving the core denoising dynamics**.

The proposed parameterized version of $L_{t-1}$ is defined as:

$$L_{t-1} = \sum_{t=2}^{T} \mathbb{E}_{q(x_t|x_0)} \left[ D_{KL} \left( q(x_{t-1}|x_t, x_0) \, \| \, p_\theta(\hat{x}_{t-1}|\hat{x}_t) \right) \right] \tag{20}$$

$L_{t-1}$ can be optimized using **stochastic timestep sampling**, where random timesteps are drawn from a uniform distribution:

$$L_{t-1} = \arg\min_\theta \mathbb{E}_{t \sim U(2,T)} D_{KL} \left( q(x_{t-1}|x_t, x_0) \, \| \, p_\theta(\hat{x}_{t-1}|\hat{x}_t) \right) \tag{21}$$

Thus, we rewrite $L_{t-1}$ as:

$$L_{t-1} = \arg\min_\theta \mathbb{E}_{t \sim U(2,T)} D_{KL} \left( \mathcal{N}(x_{t-1}; \mu_q(t), \Sigma_q(t)) \| \mathcal{N}(\hat{x}_{t-1}; \hat{\mu}_\theta, \Sigma_q(t)) \right) \tag{22}$$

where $\mu_q(t)$ and $\Sigma_q(t)$ are defined as:

$$\mu_q(t) = \frac{\sqrt{\alpha}(1 - \bar{\alpha}_{t-1})x_t + \sqrt{\bar{\alpha}}_{t-1}(1 - \alpha_t)x_0}{1 - \bar{\alpha}_t}, \tag{23}$$

$$\Sigma_q(t) = \frac{(1 - \alpha_t)(1 - \bar{\alpha}_{t-1})}{1 - \bar{\alpha}_t} I. \tag{24}$$

The goal is to model $p_\theta(\hat{x}_{t-1}|\hat{x}_t)$ such that its distribution closely approximates $q(x_{t-1}|x_t, x_0)$(*Hoet al.*, 2020). Therefore, we parameterize $p_\theta(\hat{x}_{t-1}|\hat{x}_t)$ as a Gaussian with mean $\hat{\mu}_\theta$ and variance $\Sigma_q(t)$, where $\hat{\mu}_\theta$ is defined as:

$$\hat{\mu}_\theta = \frac{\sqrt{\alpha}(1 - \bar{\alpha}_{t-1})x_\theta(\hat{x}_{t+1}, t) + \sqrt{\bar{\alpha}}_{t-1}(1 - \alpha_t)x_0}{1 - \bar{\alpha}_t}. \tag{25}$$

Here, $x_\theta(\hat{x}_{t+1}, t)$ is a neural network that predicts $x_t$, given the noisy estimate $\hat{x}_{t+1}$ and the timestep $t$. The network learns to estimate the denoised $x_t$ at each step of the reverse process.

Using this definition of $\hat{\mu}_\theta$, the loss term $L_{t-1}$ can be expressed as:

$$L_{t-1} = \arg\min_{\theta} \mathbb{E}_{t \sim U(1,T-1)} \frac{1}{2\Sigma_q(t)} \left\| \frac{\sqrt{\alpha}(1-\bar{\alpha}_{t-1})x_\theta(\hat{x}_{t+1},t) + \sqrt{\bar{\alpha}}_{t-1}(1-\alpha_t)x_0}{1-\bar{\alpha}_t} \right. $$
$$\left. - \frac{\sqrt{\alpha}(1-\bar{\alpha}_{t-1})x_t + \sqrt{\bar{\alpha}}_{t-1}(1-\alpha_t)x_0}{1-\bar{\alpha}_t} \right\|_2^2 . \tag{26}$$

Equation 26 can be further simplified (see Appendix A for the complete derivation) as:

$$L_{t-1} = \arg\min_{\theta} \mathbb{E}_{t \sim U(1,T-1)} \frac{\sqrt{\alpha}(1-\bar{\alpha}_{t-1})}{2\Sigma_q(t)(1-\bar{\alpha}_t)} \left\| \hat{x}_\theta(\hat{x}_{t+1},t) - x_t \right\|_2^2 \tag{27}$$

Optimizing $L_{t-1}$ involves training a neural network $\hat{x}_\theta(\hat{x}_{t+1},t)$ to predict $x_t$ given the estimated variable $\hat{x}_{t+1}$ and the timestep $t$. This differs from the loss in Equation 15, where the network $\hat{x}_\theta(x_t,t)$ is conditioned on the noisy input $x_t$ to predict the original noiseless input $x_0$.

To estimate a latent variable $x_t$ using Equation 27, instead of randomly sampling timesteps during training, we partition the full timestep space $[1,T]$ into sequential chunks and map each timestep to a neural network layer.

In the naive case where we use all $T$ timesteps, we would require a neural network with $N = T$ layers. This mapping creates an effective equivalence to having $N$ independent neural networks, where each layer processes a single timestep. When $N = T-1$, each timestep $t \in [1,T-1]$ in the forward process corresponds to a unique neural network layer.

However, since $T$ is typically very large (e.g., $T = 1000$), which is **much greater than** the typical number of layers in a neural network, this would be computationally infeasible. To address this, we introduce a **timestep skip parameter** $\tau > 1$, which reduces the number of layers to:

$$N = \frac{T}{\tau}. \tag{28}$$

This approach allows the data distribution to be recovered in $N$ steps—**significantly fewer than** $T$—which improves efficiency while maintaining sample quality.

Thus, the reverse process begins with white noise $x_T \sim \mathcal{N}(0,I)$, which is passed through the first layer of the network to estimate the latent variable at timestep $T - \tau$. Subsequent layers estimate the latent variables at $T - n\tau$ for $2 \leq n \leq N$. To maintain a structured mapping between timesteps and layers, we label each layer according to the timestep of the latent variable it estimates. Specifically, **layer 1 of the neural network corresponds to timestep** $l = T - \tau$ (see Figure 1). Each layer generates an intermediate estimate $\hat{x}_{T-n\tau} \in \mathbb{R}^{f \times h}$, which is then used by the next layer to produce $\hat{x}_{T-(n+1)\tau} \in \mathbb{R}^{f \times h}$.

This sequential **denoising process eliminates the need for stochastic sampling**. Moreover, the timesteps $t$ are **implicitly encoded** by the neural network layers, removing the need for explicit timestep conditioning. Thus, Equation 27 is implemented as:

$$L_{t-1} = \sum_{t=T-\tau}^{T-(N-1)\tau} \lambda_t \left\| \hat{x}_\theta^{l=t}(\hat{x}_{t+\tau}) - x_t \right\|_2^2 . \tag{29}$$

The loss term $L_{t-1}$ is optimized by training a neural network $\hat{x}_\theta(\hat{x}_{t+1},t)$ to predict $x_t$, given the latent variable estimate $\hat{x}_{t+1}$ from the previous layer and the corresponding timestep $t$. Here, $\lambda_t$ represents the contribution of layer $l = t$ to the overall loss $L_{t-1}$.

In (Ho et al., 2020), $t$ is sampled randomly, and the expectation $E_{t,x_0,\epsilon_0}[L_{t-1}]$ (Equation 17) is used to estimate the variational lower bound $L_{vlbfull}$ (Equation 19). However, the method proposed by (Ho et al.,

2020) results in samples that do not achieve competitive log-likelihoods (Nichol & Dhariwal, 2021). Log-likelihood is a key metric in generative models, driving them to capture all modes of the data distribution (Razavi et al., 2019). Inspired by this, we aim to optimize the full $L_{vlb}$ efficiently.

To compute the loss $L_0$, the output $\hat{x}_{T-(N-1)\tau}$ from layer $l = T - (N-1)\tau$ is passed to the final layer $l = T - N\tau$ of the neural network. The final predicted $\hat{x}_0$ is then given by:

$$\hat{x}_0 = \hat{x}_\theta^{l=T-N\tau}(\hat{x}_{T-(N-1)\tau}). \tag{30}$$

This prediction is used to estimate the probability $p_\theta(\hat{x}_0 \mid \hat{x}_{T-(N-1)\tau})$, which reconstructs the original indices of the input $x_0^d$ as defined by the codebook (see Figure 1).

Similar to Nichol & Dhariwal (2021), we use the cumulative distribution function (CDF) of a Gaussian distribution to estimate $p_\theta(\hat{x}_0 \mid \hat{x}_{T-(N-1)\tau})$. The loss $L_0$ is then computed as:

$$L_0 = -\log p_\theta(\hat{x}_0 \mid \hat{x}_{T-(N-1)\tau}) \tag{31}$$

The term $L_T$ is not modeled by the neural network and does not depend on $\theta$. It approaches zero if the forward noising process sufficiently corrupts the data distribution such that $q(x_T \mid x_0) \approx \mathcal{N}(0, I)$. Since $L_T$ can be computed as the KL divergence between two Gaussian distributions, the total variational loss is given by:

$$L_{vlbfull} = L_0 + L_{t-1} + L_T. \tag{32}$$

In practice, we ignore $L_T$ during implementation and compute the truncated loss as:

$$L_{vlb} = L_0 + L_{t-1}. \tag{33}$$

While $L_T$ is theoretically constant and does not depend on the model parameters, its inclusion introduces practical challenges during training. Specifically, the KL divergence term:

$$L_T = \mathrm{D_{KL}}(q(x_T|x_0)\|p(x_T)) \tag{34}$$

reflects the mismatch between the prior $p(x_T) \sim \mathcal{N}(0, I)$ and the distribution $q(x_T|x_0)$, which is influenced by the forward noising process. If the noise schedule is not perfectly tuned, this mismatch can cause $L_T$ to dominate the overall loss, leading to unstable optimization. This phenomenon has been observed empirically and aligns with findings in prior work (Nichol & Dhariwal, 2021).

Additionally, although $L_T$ remains constant with respect to model parameters, it can distort the total loss magnitude, overshadowing model-dependent terms such as $L_0$ and $L_{t-1}$. This distortion may hinder convergence and lead to suboptimal optimization dynamics (see Figure 3). While (Nichol & Dhariwal, 2021) proposed refining the noise schedule to mitigate this issue, we chose to exclude $L_T$ from the training loss entirely. This simplifies implementation and allows the optimization process to focus on model-dependent terms without sacrificing theoretical consistency during evaluation. Figure 2 visualizes spectrograms of audio generated at different layers of the neural network, illustrating how noise is progressively removed.

The proposed sequential denoising approach significantly reduces computational complexity by distributing the denoising process across network layers instead of requiring per-step noise resampling. This design offers three key advantages:

- **Improved stability** – Direct estimation of $\hat{x}_{t-1}$ minimizes stochasticity, leading to smoother training dynamics.

- **Faster inference** – Eliminating explicit noise sampling reduces redundant computations, accelerating generation speed.

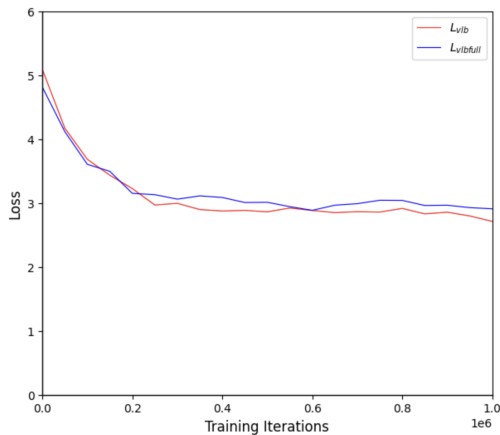

Figure 3: Learning curves comparing the full objective $L_{vlbfull}$ and $L_{vlb}$ on the LJSpeech dataset.

- **Maintained accuracy** – The hierarchical structure ensures that $\hat{x}_{t-1}$ and $\hat{x}_t$ retain sufficient information for precise reconstruction.

Algorithms 1 and 2 summarize the training and sampling procedures of the proposed method.

---

**Algorithm 1** Training Algorithm with $\tau$, $T$, $x_0$, Codebook $\mathcal{Z}$

---

1: Initialize $N = \frac{T}{\tau}$
2: **repeat**
3: Initialize $L_{t-1} = 0$
4: Map $x_0$ rows to codebook indices: $x_0^d$
5: Set $\lambda_t = 0.001$
6: Initialize $\hat{x}_{t+\tau} = x_T \sim \mathcal{N}(0, I)$
7: Sample noise: $\epsilon_0 \sim \mathcal{N}(0, I)$
8: **for** $t = T - \tau$ to $T - (N-1)\tau$:
9:    Update loss: $L_{t-1} + = \lambda_t \left\| x_\theta^{l=t}(\hat{x}_{t+\tau}) - \sqrt{\bar{\alpha}_t} x_0 + \sqrt{1 - \bar{\alpha}_t} \epsilon_0 \right\|_2^2$
10:    Update prediction: $\hat{x}_{t+\tau} = x_\theta^{l=t}(\hat{x}_{t+\tau})$
11:    Increment weight: $\lambda_t + = 0.001$
12:   **if** $t = T - (N-1)\tau$:
13:      Predict $\hat{x}_0 = x_\theta^{l=t-\tau}(\hat{x}_{t+\tau})$
14:      Compute likelihood $p(\hat{x}_0 \mid \hat{x}_t)$ for restoring $x_0^d$
15:      Compute loss: $L_0 = -\log p(\hat{x}_0 \mid \hat{x}_t)$
16: Compute total loss: $L_{vlb} = L_0 + L_{t-1}$
17: Update the neural networks $x_\theta^l(.)$ to minimize $L_{vlb}$
18: **until** $L_{vlb}$ converges

---

**Algorithm 2** Sampling Algorithm with $\tau$, $x_t$, $x_\theta^{l=t}(.)$, and $T - \tau \leq t \leq T - N\tau$

---

1: Initialize $\hat{x}_{t+\tau} = x_T \sim \mathcal{N}(0, I)$
2: **for** $t = T - \tau$ to $T - N\tau$:
3:    Update estimate: $x_t = x_\theta^{l=t}(\hat{x}_{t+\tau})$
4:    Update prediction: $\hat{x}_{t+\tau} = x_\theta^{l=t}(\hat{x}_{t+\tau})$
5: **end for**
6: Return final prediction: $x_{T-N\tau} = x_0$

---

## 4.2 Conditional Speech Generation

To enable the model to generate speech conditioned on specific acoustic features, we modify the neural network layer to incorporate these features, denoted as $y$. The loss function is now defined as:

$$L_{t-1} = \sum_{t=T-\tau}^{T-(N-1)\tau} \lambda_t \left\| \hat{x}_\theta^{l=t}(\hat{x}_{t+\tau}, y) - x_t \right\|_2^2 \tag{35}$$

We design the score network $\hat{x}_\theta^l(.,.)$ to process both the estimated value $\hat{x}_{t+\tau}$ and the acoustic features $y$. To achieve this, we use feature-wise linear modulation (FiLM) (Perez et al., 2018), as applied in (Chen et al., 2020).

FiLM takes the Mel spectrogram $y$ as input and adaptively learns transformation functions $f(y)$ and $h(y)$ to output the modulation parameters $\gamma$ and $\beta$:

$$\gamma = f(y), \quad \beta = h(y)$$

These parameters modulate the intermediate layer activations using an affine transformation, applied element-wise as:

$$FiLM(\hat{x}_{t+\tau}) = \gamma \odot \hat{x}_{t+\tau} + \beta \tag{36}$$

Here, both $\gamma$ and $\beta \in \mathbb{R}^{f \times h}$ are learnable transformations of $y$, and $\odot$ represents the Hadamard (element-wise) product. This conditioning ensures that the network adapts its intermediate representations dynamically based on the Mel spectrogram features.

To compute $L_0$ for conditional generation, we first estimate the conditional probability $p_\theta(\hat{x}_0 \mid \hat{x}_{T-(N-1)\tau}, y)$, which predicts the original indices $x_0^d$ of the input $x_0$ as established by the codebook. The loss $L_0$ is then computed as:

$$L_0 = -\log p_\theta(\hat{x}_0 \mid \hat{x}_{T-(N-1)\tau}, y) \tag{37}$$

## 5 Alternative Loss Functions

To improve the quality of generated speech samples, we explored alternative objective functions. These loss functions aim to balance simplicity with performance, focusing on generating clearer and more accurate speech.

The first alternative, shown in Equation 37, is a simplified version of the original loss in Equation 15,. This loss minimizes the difference between the predicted output $\hat{x}_\theta$ and the original input $x_0$, making it more computationally efficient.

$$L_{simple} = \left\| \hat{x}_\theta^{l=T-(N-1)\tau}(\hat{x}_{t+\tau}) - x_0 \right\|_2^2 \tag{38}$$

The second alternative, shown in Equation 38, is a hybrid loss that combines the simplicity of $L_{simple}$ with the full variational lower bound loss $L_{vlbfull}$ from (Nichol & Dhariwal, 2021). This hybrid approach aims to leverage the benefits of both simplified and complete loss functions to improve model performance across various conditions.

$$L_{hybrid} = L_{simple} + \lambda L_{vlbfull} \tag{39}$$

where $L_{vlbfull} = L_0 + L_{t-1} + L_T$.

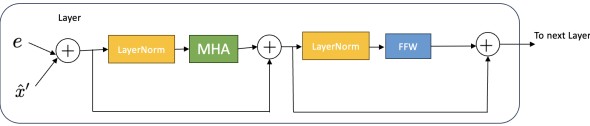

Figure 4: A single layer of the transformer model used for data recovery (see Figure 4).

## 6 Models

### 6.1 Encoder

The encoder (used in the forward process, see Figure 1) consists of a single layer of 256 convolutional filters with a kernel size of 16 samples and a stride of 8 samples. This configuration is chosen to capture sufficient temporal and spectral information from the input speech. The encoder generates a representation $x_0 \in \mathbb{R}^{F \times T'}$, where $F$ is the feature dimension and $T'$ is the time axis.

$$x_0 = ReLU(\text{conv1d}(x))$$

The latent variables $x_t \in \mathbb{R}^{F \times T'}$ are then generated from $x_0$ during the forward diffusion process. To reconstruct the original signal, we use a transposed convolutional layer at the end of the reverse process, which has the same stride and kernel size as the encoder to ensure symmetry.

### 6.2 Data Recovery Model

For data recovery, an estimate $\hat{x}_{T-(n-1)\tau} \in \mathbb{R}^{F \times T'}$ is first normalized using layer normalization. This normalized estimate is then passed through a linear layer with a dimension of $F$.

Next, the output is chunked into segments of size $s$ along the $T'$ axis with a 50% overlap to better capture temporal dependencies. The chunked output $\hat{x}'_{T-(n-1)\tau} \in \mathbb{R}^{F \times s \times V}$, where $V$ represents the total number of chunks, is passed through a neural network layer (Figure 4).

Each layer of this network is a transformer with 8 attention heads and a 768-dimensional feedforward network. The transformer processes the input $\hat{x}'_{T-(n-1)\tau} \in \mathbb{R}^{F \times s \times V}$ and outputs $\hat{x}'_{T-n\tau} \in \mathbb{R}^{F \times s \times V}$, which is passed to the next layer $T-(n+1)\tau$. After processing, the final estimate $\hat{x}_{T-n\tau} \in \mathbb{R}^{F \times T'}$ is obtained by merging the last two dimensions of $\hat{x}'_{T-n\tau} \in \mathbb{R}^{F \times s \times V}$.

## 7 Evaluation

This section discusses the datasets and training parameters used to develop and evaluate the proposed technique, referred to as **UDPNet** (Unrolling Diffusion Process Network).

### 7.1 Dataset

To ensure comparability with existing tools and maintain alignment with trends in the speech synthesis domain, we evaluated UDPNet on two popular datasets: LJSpeech for single-speaker speech generation and VCTK for multi-speaker evaluation.

The **LJSpeech** dataset consists of 13,100 audio clips sampled at 22 kHz, totaling approximately 24 hours of single-speaker audio. Clip lengths range from 1 to 10 seconds, and all clips feature a single female speaker. Following (Chen et al., 2020), we used 12,764 utterances (23 hours) for training and 130 utterances for testing.

For multi-speaker evaluation, we used the **VCTK** dataset, which includes recordings of 109 English speakers with diverse accents, originally sampled at 48 kHz and downsampled to 22 kHz for consistency. Following

(Lam et al., 2022), we used a split where 100 speakers were used for training and 9 speakers were held out for evaluation.

**Feature Extraction:** Mel-spectrograms were extracted from each audio clip, resulting in 128-dimensional feature vectors. The extraction process used a 50-ms Hanning window, a 12.5-ms frame shift, and a 2048-point FFT, with frequency limits of 20 Hz (lower) and 12 kHz (upper), similar to (Chen et al., 2020).

## 7.2 Training Parameters

UDPNet was trained on a single NVIDIA V100 GPU using the Adam optimizer. A cyclical learning rate (Smith, 2017) was employed, with the learning rate varying between $1e - 4$ and $1e - 1$. The batch size was set to 32, and training was performed over 1 million steps, taking approximately 8 days to complete.

For **conditional speech generation**, Mel-spectrograms extracted from ground truth audio were used as conditioning features during training. During testing, Mel-spectrograms were generated by Tacotron 2 (Shen et al., 2018). To generate the FiLM parameters $\beta$ and $\gamma$, we adopted the upsampling block approach proposed in Chen et al. (2020), where these parameters modulate layer activations to incorporate conditioning information.

**Layer Contribution to Loss:** Each neural network layer's contribution to the total loss $L_{t-1}$ (Equation 20) was weighted using a layer-specific factor $\lambda$. The weights were initialized at $\lambda = 0.001$ for the first layer and incremented by 0.001 for each subsequent layer. This approach ensured that higher layers, which handle progressively refined denoising steps, had a greater impact on the overall loss. This weighting helps the model prioritize more challenging denoising tasks, which are typically assigned to the higher layers.

## 7.3 Baseline Models and Metrics

To evaluate UDPNet, we compared its performance against several state-of-the-art vocoders with publicly available implementations. The selected baseline models are:

- WaveNet (Oord et al., 2016) [1]

- WaveGlow (Prenger et al., 2018) [2]

- MelGAN (Kumar et al., 2019) [3]

- HiFi-GAN (Kong et al., 2020a) [4]

- WaveGrad (Chen et al., 2020) [5]

- DiffWave (Kong et al., 2020b) [6]

- BDDM (Lam et al., 2022) [7]

- FastDiff (Huang et al., 2022a) [8]

- CoMoSpeech (Li et al., 2024)[9]

We assessed the models using a combination of subjective and objective metrics:

---

[1] `https://github.com/r9y9/wavenet_vocoder`
[2] `https://github.com/NVIDIA/waveglow`
[3] `https://github.com/descriptinc/melgan-neurips`
[4] `https://github.com/jik876/hifi-gan`
[5] `https://github.com/tencent-ailab/bddm`
[6] `https://github.com/tencent-ailab/bddm`
[7] `https://github.com/tencent-ailab/bddm`
[8] `https://FastDiff.github.io/`
[9] https://github.com/zhenye234/CoMoSpeech

- **Mean Opinion Score (MOS):** Human evaluators rated the naturalness and quality of generated speech on a 5-point scale (1 = Bad, 5 = Excellent). Evaluators were recruited via Amazon Mechanical Turk, wore headphones, and rated 10 samples each.

- **Objective MOS Prediction:** We used three deep learning-based MOS prediction tools: SSL-MOS [10] (Cooper et al., 2022), MOSA-Net [11] (Zezario et al., 2022), and LDNet [12] (Huang et al., 2022c). These tools are widely used in the VoiceMOS challenge (Huang et al., 2022b).

- **$F_0$ Frame Error (FFE):** This metric measures pitch accuracy by quantifying discrepancies between the generated and ground truth audio.

**Objective MOS Prediction Tools:** SSL-MOS is a Wav2Vec-based model fine-tuned for MOS prediction by adding a linear layer to the Wav2Vec backbone. MOSA-Net incorporates cross-domain features, including spectrograms, raw waveforms, and features from self-supervised learning speech models, to enhance its predictions. LDNet estimates listener-specific MOS scores and averages them across all listeners for a final score.

## 7.4 Model Configurations

UDPNet was evaluated using different forward diffusion steps (**fsteps**) while maintaining a fixed number of 8 reverse steps. The forward steps considered were 1200, 1000, 960, 720, and 240, corresponding to skip parameters $\tau = \{150, 125, 120, 90, 30\}$, respectively. Each configuration accepted a 0.3-second audio input.

The choice of 1200 steps was motivated by our desire to explore different diffusion step configurations while ensuring a consistent number of 8 reverse steps across all models. This setup allows us to systematically analyze how the number of forward steps impacts both synthesis quality and computational efficiency.

To ensure a fair comparison with other models, such as WaveGrad, which operates with 1000 forward steps, we also evaluated UDPNet using 1000 forward steps while maintaining 8 reverse steps.

The forward noise schedule $\alpha_i$ was defined as a linear progression across all steps:

$$\alpha_i = Linear(\alpha_1, \alpha_N, N),$$

where $N$ represents the total number of forward steps. For example, with 1200 forward steps, the schedule was specified as $Linear(1 \times 10^{-4}, 0.005, 1200)$.

During training, we conditioned UDPNet on Mel-spectrograms extracted from ground truth audio, while for testing, spectrograms generated by Tacotron 2 (Shen et al., 2018) were used. To enhance conditional generation, FiLM parameters $\beta$ and $\gamma$ were generated following the upsampling block approach proposed in Chen et al. (2020), modulating the activations of corresponding layers.

To ensure that higher layers, which handle finer denoising tasks, had a greater influence during training, each layer's contribution to the loss $L_{t-1}$ was weighted. The weights $\lambda$ were initialized at 0.001 for the first layer and incremented by 0.001 for each subsequent layer. This design ensures a progressive emphasis on higher layers, aligning with their role in refining the denoised output.

## 7.5 Results

### 7.5.1 Gradient Noise Scales of the Objective Functions

We evaluated the gradient noise scales of three proposed objective functions: $L_{vlb}$, $L_{simple}$, and $L_{hybrid}$, following the methodology in (McCandlish et al., 2018) and (Nichol & Dhariwal, 2021). The models were trained on the LJSpeech dataset for single-speaker evaluation, using a configuration of 1200 forward steps and 8 reverse steps, which offered a balance between computational efficiency and performance.

---

[10] https://github.com/nii-yamagishilab/mos-finetune-ssl
[11] https://github.com/dhimasryan/MOSA-Net-Cross-Domain
[12] https://github.com/unilight/LDNet

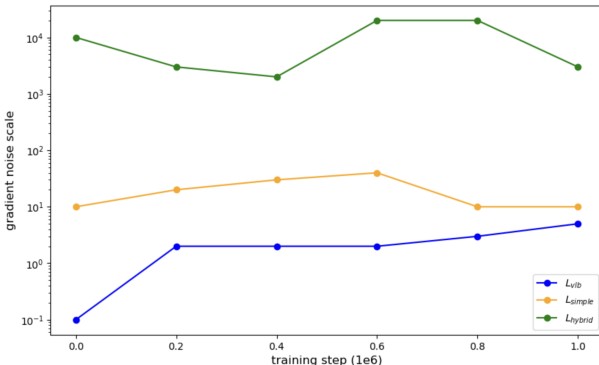

Figure 5: Gradient noise scales for the $L_{vlb}$, $L_{hybrid}$, and $L_{simple}$ objectives on the LJSpeech dataset (see Figure 5).

As shown in Figure 5, $L_{hybrid}$ exhibited the highest gradient noise levels. This behavior is attributed to the inclusion of the $L_T$ term in the objective function. As noted by (Nichol & Dhariwal, 2021), $L_T$, which represents the KL divergence between the prior and forward noising process at the final timestep, introduces significant noise during uniform timestep sampling. This makes training less stable and impairs convergence.

In contrast, $L_{vlb}$ demonstrated more stable gradient behavior compared to both $L_{simple}$ and $L_{hybrid}$, making it the preferred choice for subsequent evaluations. Its stability ensures smoother optimization and better performance during training.

### 7.5.2 Single-Speaker Evaluation

To assess UDPNet's performance in conditional speech generation on a single-speaker dataset, we conducted a comprehensive evaluation using both subjective and objective metrics. Table 1 presents the Mean Opinion Score (MOS), objective MOS estimates from SSL-MOS, MOSA-Net, and LDNet, as well as the Real-Time Factor (RTF), which measures inference speed. The best-performing configuration of UDPNet, using 1200 forward steps and 8 reverse steps (**1200, 8**), achieved a subjective MOS of 4.49 just 0.23 points below the ground truth (4.72), demonstrating its ability to produce highly natural and perceptually accurate speech. It also outperformed all baseline models across objective metrics, achieving the highest scores on SSL-MOS, MOSA-Net, and LDNet, and the lowest $F_0$ Frame Error (FFE) of 2.3%, indicating superior pitch preservation and low distortion. To enable fair comparison with other diffusion-based models using 1000 forward steps, we evaluated UDPNet under the same setting (**1000, 8**), which yielded a subjective MOS of 4.44—only 0.05 lower than the best configuration—while maintaining strong performance across all objective metrics. Moreover, this setup achieved a significantly lower RTF of 0.00389 compared to 38.2 for WaveGrad, highlighting the substantial efficiency gains enabled by UDPNet's layered design.

We observed that increasing the number of forward steps generally improved speech quality, as reflected in both subjective and objective MOS scores. This improvement is likely due to more gradual and fine-grained denoising across layers. However, the marginal gain beyond 1000 steps suggests diminishing returns in quality. At the same time, reducing the number of forward steps led to faster generation. The configuration with 240 forward steps achieved the lowest RTF of 0.00182—demonstrating that UDPNet can scale toward ultra-fast inference with only a modest reduction in perceptual quality. Additionally, our choice to predict latent variables $x_t$ in the variational objective $L_{\text{vlb}}$, rather than directly predicting $x_0$, appears to reduce cumulative prediction error. Prior work (Zhou et al., 2023) has shown that such errors contribute significantly to speech distortion, and our results suggest this design decision contributes meaningfully to UDPNet's quality. We also included CoMoSpeech, a recent consistency model-based speech synthesizer that generates audio in a single step. CoMoSpeech achieves faster inference with an RTF of 0.0058—approximately 2.45 times faster than UDPNet (**1200, 8**). However, this speed comes at the expense of quality: UDPNet surpasses CoMoSpeech by +0.31 in subjective MOS and outperforms it across all objective metrics. These results highlight the strength of UDPNet's progressive denoising approach, which yields higher fidelity and better

pitch accuracy while maintaining low-latency inference. Unlike CoMoSpeech, UDPNet does not rely on a pre-trained teacher model or distillation, simplifying training without compromising output quality.

This comparison illustrates a key trade-off in generative speech synthesis: while CoMoSpeech offers extremely fast inference via single-step generation, UDPNet's layered refinement approach achieves superior perceptual quality, making it more suitable for applications where audio fidelity is critical. Notably, even at its highest-quality configuration, UDPNet remains orders of magnitude faster than traditional diffusion models like WaveGrad, striking an effective balance between quality and efficiency.

Table 1: Evaluation results of UDPNet compared to state-of-the-art tools on the LJSpeech test dataset. Metrics include subjective MOS, objective MOS (SSL-MOS, MOSA-Net, and LDNet), $F_0$ Frame Error (FFE), and Real-Time Factor (RTF).

| LJSpeech Test Dataset | | | | | | |
|---|---|---|---|---|---|---|
| Model | MOS($\uparrow$) | SSL-MOS($\uparrow$) | MOSA-Net($\uparrow$) | LDNet($\uparrow$) | FFE($\downarrow$) | RTF($\downarrow$) |
| Ground Truth | 4.72±0.15 | 4.56 | 4.51 | 4.67 | - | - |
| BDDM (12 steps) | 4.38±0.15 | 4.23 | 4.17 | 4.42 | 3.6% | 0.543 |
| DiffWave (200 steps) | 4.43±0.13 | 4.31 | 4.28 | 4.36 | 2.6% | 5.9 |
| WaveGrad (1000 steps) | 4.32±0.15 | 4.27 | 4.23 | 4.31 | 2.8% | 38.2 |
| HIFI-GAN | 4.26±0.14 | 4.19 | 4.13 | 4.27 | 3.3% | 0.0134 |
| MelGAN | 3.49±0.12 | 3.33 | 3.27 | 3.42 | 6.7% | 0.00396 |
| WaveGlow | 3.17±0.14 | 3.12 | 3.09 | 3.14 | 7.3% | 0.0198 |
| WaveNet | 3.61±0.15 | 3.51 | 3.47 | 3.54 | 6.3% | 318.6 |
| CoMoSpeech | 4.18±0.15 | 4.13 | 4.05 | 4.17 | 3.1% | 0.0058 |
| UDPNet (fsteps: 1200, rsteps: 8) | **4.49**±0.12 | **4.43** | **4.35** | **4.44** | **2.3%** | 0.0142 |
| UDPNet (fsteps: 1000, rsteps: 8) | 4.44±0.12 | 4.37 | 4.31 | 4.40 | 3.3% | 0.00389 |
| UDPNet (fsteps: 960, rsteps: 8) | 4.33±0.15 | 4.283 | 4.23 | 4.31 | 3.7% | 0.00371 |
| UDPNet (fsteps: 720, rsteps: 8) | 4.17±0.15 | 4.12 | 4.09 | 4.14 | 4.3% | 0.002912 |
| UDPNet (fsteps: 240, rsteps: 8) | 4.09±0.13 | 4.05 | 4.01 | 4.05 | 4.7% | 0.00182 |

### 7.5.3 Multi-Speaker Evaluation

We evaluated UDPNet on the multi-speaker VCTK dataset to assess its generalization to unseen speakers. As shown in Table 2, the best configuration, UDPNet (1200, 8), achieved a subjective MOS of 4.38—closely matching DiffWave and just 0.25 below the ground truth (4.63). Despite DiffWave attaining a slightly lower $F_0$ Frame Error (3.2% vs. 3.3%), UDPNet outperformed all baseline models across objective MOS metrics, including SSL-MOS, MOSA-Net, and LDNet, indicating its ability to produce high-quality, intelligible speech. To ensure fairness, we also evaluated UDPNet (1000, 8), which yielded a subjective MOS of 4.35—surpassing WaveGrad (4.26)—while reducing inference time dramatically (RTF 0.00389 vs. 38.2). This demonstrates that UDPNet maintains strong performance even with fewer forward steps. Although UDPNet (1200, 8) is slower than CoMoSpeech (RTF 0.0142 vs. 0.0058), it provides significantly better quality (MOS 4.38 vs. 4.08) and higher scores across all objective metrics. This reflects the strength of UDPNet's progressive denoising architecture in delivering superior fidelity, even when not operating in single-step inference mode.

The design of UDPNet—including its fixed 8-step reverse process and novel layer-timestep mapping—enables efficient denoising without compromising quality. This flexibility is particularly evident in lower-step configurations. For instance, UDPNet (240, 8) achieves the fastest RTF of 0.00182 while still maintaining a subjective MOS of 4.04, highlighting its adaptability across latency-constrained environments.In summary, UDPNet demonstrates strong generalization in multi-speaker synthesis, achieving high perceptual quality, competitive pitch accuracy, and scalable inference speeds. Its configurable architecture enables a favorable balance between efficiency and naturalness, making it suitable for real-world applications such as multi-speaker transcription, voice assistants, and conversational agents.

### 7.5.4 Unconditional Speech Generation

To evaluate the capability of UDPNet in unconditional speech generation, we trained the model on the multi-speaker VCTK dataset. Speech samples were generated by sampling random white noise and passing it through the trained UDPNet without conditioning on any acoustic features. The results are summarized in Table 3.

Table 2: Evaluation results of the conditioned version of the proposed method compared to state-of-the-art tools on the evaluation metrics using the multi-speaker dataset (VCTK).

| VCTK Test Dataset | | | | | | |
|---|---|---|---|---|---|---|
| **Model** | **MOS(↑)** | **SSL-MOS(↑)** | **MOSANet(↑)** | **LDNet(↑)** | **FFE(↓)** | **RTF(↓)** |
| Ground Truth | 4.63±0.05 | 4.57 | 4.69 | 4.65 | - | - |
| BDDM (12 steps) | 4.33±0.05 | 4.28 | 4.25 | 4.35 | 4.3% | 0.543 |
| DiffWave (200 steps) | **4.38**±0.03 | 4.41 | 4.32 | 4.33 | **3.2%** | 5.9 |
| WaveGrad (1000 steps) | 4.26±0.05 | 4.31 | 4.21 | 4.24 | 3.4% | 38.2 |
| HIFI-GAN | 4.19±0.14 | 4.12 | 4.16 | 4.18 | 3.9% | 0.0134 |
| MelGAN | 3.33±0.05 | 3.27 | 3.24 | 3.37 | 7.7% | 0.00396 |
| WaveGlow | 3.13±0.05 | 3.12 | 3.16 | 3.09 | 8.2% | 0.0198 |
| WaveNet | 3.53±0.05 | 3.43 | 3.45 | 3.46 | 7.2% | 318.6 |
| CoMoSpeech | 4.08±0.05 | 4.01 | 4.06 | 4.10 | 3.7% | 0.0058 |
| UDPNet (fsteps: 1200, rsteps: 8) | **4.38**±0.12 | **4.43** | **4.36** | **4.40** | 3.3% | 0.0142 |
| UDPNet (fsteps: 1000, rsteps: 8) | 4.35±0.05 | 4.39 | 4.36 | 4.37 | 3.3% | 0.00389 |
| UDPNet (fsteps: 960, rsteps: 8) | 4.28±0.05 | 4.23 | 4.25 | 4.29 | 4.2% | 0.00371 |
| UDPNet (fsteps: 720, rsteps: 8) | 4.12±0.05 | 4.11 | 4.13 | 4.08 | 4.6% | 0.002912 |
| UDPNet (fsteps: 240, rsteps: 8) | 4.04±0.03 | 4.01 | 3.91 | 4.01 | 5.2% | **0.00182** |

Among the tested configurations, the best-performing model, **UDPNet (fsteps: 1200, rsteps: 8)**, achieved a subjective MOS of 3.11. Notably, while the generated speech initially sounds natural and coherent, the intelligibility tends to degrade over longer durations. This suggests that the model struggles with maintaining temporal coherence, an issue that could be explored in future work to improve long-form speech generation.

Despite this limitation, UDPNet demonstrates strong performance in generating clean speech with minimal noise or artifacts. The model maintains a favorable trade-off between quality and efficiency, as reflected in the real-time factor (RTF) across different configurations. In particular, **the fastest configuration, UDPNet (fsteps: 240, rsteps: 8), achieved an RTF of 0.00162**, highlighting the scalability and computational efficiency of the proposed approach.

Table 3: Evaluation of UDPNet for unconditional speech generation on the VCTK test dataset. We report subjective MOS, objective MOS (SSL-MOS, MOSA-Net, LDNet), and real-time factor (RTF).

| VCTK Test Dataset | | | | | |
|---|---|---|---|---|---|
| **Model** | **MOS (↑)** | **SSL-MOS (↑)** | **MOSA-Net (↑)** | **LDNet (↑)** | **RTF (↓)** |
| BDDM (12 steps) | 3.33±0.05 | 3.26 | 3.25 | 3.32 | 0.543 |
| DiffWave (200 steps) | 3.28±0.03 | 3.31 | 3.32 | 3.33 | 5.9 |
| WaveGrad (1000 steps) | 3.26±0.05 | 3.31 | 3.21 | 3.24 | 38.2 |
| HIFI-GAN | 3.19±0.14 | 3.12 | 3.16 | 3.18 | 0.0134 |
| MelGAN | 3.33±0.05 | 3.27 | 3.24 | 3.37 | 0.00396 |
| WaveGlow | 3.13±0.05 | 3.12 | 3.16 | 3.09 | 0.0198 |
| WaveNet | 3.11±0.05 | 3.09 | 3.23 | 3.12 | 0.0058 |
| UDPNet (fsteps: 1200, rsteps: 8) | 3.11±0.12 | 3.17 | 3.16 | 3.23 | 0.0038 |
| UDPNet (fsteps: 1000, rsteps: 8) | 3.07±0.12 | 3.15 | 3.13 | 3.18 | 0.00367 |
| UDPNet (fsteps: 960, rsteps: 8) | 3.04±0.05 | 3.09 | 3.01 | 3.09 | 0.00351 |
| UDPNet (fsteps: 720, rsteps: 8) | 3.02±0.05 | 3.07 | 3.01 | 3.06 | 0.002812 |
| UDPNet (fsteps: 240, rsteps: 8) | 2.98±0.03 | 3.02 | 3.03 | 3.08 | 0.00162 |

## 8 Conclusion

In this paper, we introduced UDPNet, a novel approach for accelerating speech generation in diffusion models by leveraging the structure of neural network layers. By progressively recovering the data distribution from white noise, each neural network layer performs implicit denoising. Through the use of a skip parameter $\tau$, we effectively map neural network layers to the forward diffusion process, reducing the number of recovery steps required and improving efficiency.

Our modified objective function allows the model to balance accuracy and speed, and we further enhanced conditional speech generation by incorporating Feature-wise Linear Modulation (FiLM) to integrate acoustic features into the denoising process. Through extensive evaluations on both single-speaker and multi-speaker

datasets, UDPNet demonstrated the ability to produce high-quality speech samples while maintaining competitive generation speed.

While the results are promising, future work could explore the impact of increasing the number of forward steps, investigate the coherence degradation over time in unconditional speech generation, and apply UDPNet to additional speech tasks or other generative modeling domains. Overall, UDPNet offers a significant step forward in efficient and high-quality speech generation for diffusion-based models.

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

## Impact Statement

"This paper presents work whose goal is to advance the field of Machine Learning. There are many potential societal consequences of our work, none which we feel must be specifically highlighted here.

## A   Appendix.

## B   Derivation of Equation 27

We begin by defining the loss function $L_{t-1}$ as the Kullback-Leibler (KL) divergence between the true posterior distribution $q(x_{t-1}|x_t, x_0)$ and the learned model distribution $p_\theta(\hat{x}_{t-1}|\hat{x}_t)$:

$$L_{t-1} = \arg\min_\theta \mathbb{E}_{t \sim U(2,T)} D_{\text{KL}} \left( q(x_{t-1}|x_t, x_0) \| p_\theta(\hat{x}_{t-1}|\hat{x}_t) \right). \tag{40}$$

Assuming that both distributions are Gaussian, the KL divergence can be rewritten as:

$$L_{t-1} = \arg\min_\theta \mathbb{E}_{t \sim U(2,T)} D_{\text{KL}} \left( \mathcal{N}(x_{t-1}; \mu_q(t), \Sigma_q(t)) \| \mathcal{N}(\hat{x}_{t-1}; \hat{\mu}_\theta, \Sigma_q(t)) \right), \tag{41}$$

where the covariance matrix $\Sigma_q(t)$ and the means $\mu_q(t)$ and $\hat{\mu}_\theta$ are given by:

$$\Sigma_q(t) = \frac{(1 - \alpha_t)(1 - \bar{\alpha}_{t-1})}{1 - \bar{\alpha}_t} I, \tag{42}$$

$$\mu_q(t) = \frac{\sqrt{\alpha}(1 - \bar{\alpha}_{t-1})x_t + \sqrt{\bar{\alpha}_{t-1}}(1 - \alpha_t)x_0}{1 - \bar{\alpha}_t}, \tag{43}$$

$$\hat{\mu}_\theta = \frac{\sqrt{\alpha}(1 - \bar{\alpha}_{t-1})x_\theta(\hat{x}_{t+1}, t) + \sqrt{\bar{\alpha}_{t-1}}(1 - \alpha_t)x_0}{1 - \bar{\alpha}_t}. \tag{44}$$

Since both distributions share the same covariance $\Sigma_q(t)$, the KL divergence simplifies to:

$$D_{\text{KL}} \left( \mathcal{N}(\mu_q(t), \Sigma_q(t)) \| \mathcal{N}(\hat{\mu}_\theta, \Sigma_q(t)) \right) = \frac{1}{2} \left( \hat{\mu}_\theta - \mu_q(t) \right)^T \Sigma_q(t)^{-1} \left( \hat{\mu}_\theta - \mu_q(t) \right). \tag{45}$$

$$\frac{1}{2\Sigma_q(t)} \left[ \left( \frac{\sqrt{\alpha}(1 - \bar{\alpha}_{t-1})x_\theta(\hat{x}_{t+1}, t) + \sqrt{\bar{\alpha}_{t-1}}(1 - \alpha_t)x_0}{1 - \bar{\alpha}_t} - \frac{\sqrt{\alpha}(1 - \bar{\alpha}_{t-1})x_t + \sqrt{\bar{\alpha}_{t-1}}(1 - \alpha_t)x_0}{1 - \bar{\alpha}_t} \right)^2 \right]. \tag{46}$$

Since the $x_0$-dependent terms cancel out, we obtain:

$$L_{t-1} = \frac{\sqrt{\alpha}(1 - \bar{\alpha}_{t-1})}{2\Sigma_q(t)(1 - \bar{\alpha}_t)} \|x_\theta(\hat{x}_{t+1}, t) - x_t\|_2^2. \tag{47}$$

Thus, the final loss function penalizes the squared Euclidean distance between the predicted $x_t$ and the ground truth $x_t$, weighted by a scaling factor derived from the diffusion process parameters.

## C   Effect of $\tau$ on Speech Quality and Inference Time

To evaluate the impact of the skip parameter $\tau$ on both speech quality and inference speed, we fix the number of forward diffusion steps at $T = 1000$ and vary $\tau$ across $\{50, 100, 125, 200, 250, 500, 1000\}$. Table 4 presents the results on the LJSpeech test dataset, assessing both subjective and objective quality metrics.

When $\tau = 1000$, the model consists of a **single layer**, which must directly predict $x_0$ from $x_T$, skipping intermediate refinement steps. In contrast, for smaller values of $\tau$, denoising is distributed across multiple layers, allowing the model to progressively refine its predictions.

The results indicate that decreasing $\tau$ (i.e., increasing the number of reverse steps) **improves speech quality** but comes at the expense of increased inference time. This suggests that the model benefits from **progressively refining latent variables** rather than making large jumps in denoising.

However, the improvements from additional denoising steps **diminish beyond a certain point**. For example, reducing $\tau$ from 500 to 250 improves MOS from 3.17 to 4.29. However, further reducing $\tau$ from 100 to 50 results in a slight drop in MOS from 4.46 to 4.44, suggesting a possible over-smoothing effect. Beyond this saturation point, additional denoising steps yield negligible quality gains while significantly increasing inference time.

The selected $\tau$ values were chosen to cover a **range of trade-offs** between inference speed and synthesis quality. **Larger values ($\tau = 500, 1000$)** simulate near-instantaneous generation, while **smaller values ($\tau = 100, 200$)** provide high-fidelity synthesis at the cost of longer computation.

Table 4: Impact of Skip Parameter $\tau$ on Speech Quality and Inference Speed. We report subjective MOS, objective MOS (SSL-MOS, MOSA-Net, and LDNet), $F_0$ Frame Error (FFE), and Real-Time Factor (RTF) on the LJSpeech test dataset.

| LJSpeech Test Dataset | | | | | | |
|---|---|---|---|---|---|---|
| Model | MOS($\uparrow$) | SSL-MOS($\uparrow$) | MOSA-Net($\uparrow$) | LDNet($\uparrow$) | $F_0$ Frame Error (FFE)($\downarrow$) | Real-Time Factor (RTF)($\downarrow$) |
| UDPNet (fsteps: 1000, rsteps: 20, $\tau = 50$) | 4.44 ±0.15 | **4.38** | 4.36 | 4.39 | 3.1% | 0.0254 |
| UDPNet (fsteps: 1000, rsteps: 10, $\tau = 100$) | **4.46**±0.12 | 4.37 | 4.34 | **4.41** | **2.9%** | 0.0054 |
| UDPNet (fsteps: 1000, rsteps: 8, $\tau = 125$) | 4.44±0.12 | 4.37 | 4.31 | 4.40 | 3.3% | 0.00389 |
| UDPNet (fsteps: 1000, rsteps: 5, $\tau = 200$) | 4.29±0.15 | 4.19 | 4.20 | 4.25 | 4.1% | 0.00323 |
| UDPNet (fsteps: 1000, rsteps: 4, $\tau = 250$) | 3.27±0.15 | 3.74 | 3.18 | 3.22 | 4.3% | 0.00291 |
| UDPNet (fsteps: 1000, rsteps: 2, $\tau = 500$) | 3.17±0.15 | 3.12 | 3.09 | 3.09 | 6.3% | 0.001112 |
| UDPNet (fsteps: 1000, rsteps: 1, $\tau = 1000$) | 2.09±0.13 | 2.05 | 2.11 | 2.05 | 8.7% | **0.00082** |

## C.1  Conclusion

