# OpenReview forum: "Speech Synthesis By Unrolling  Diffusion Process using Neural Network Layers"
_TMLR — Accepted by TMLR_

### Review · Reviewer_cjtK · 2024-11-22

**Summary Of Contributions:**

This paper presents a speech synthesis method based on diffusion models. Specifically, the authors propose to “unroll” the reverse diffusion process to neural network layers and map each network layer to several diffusion steps, with the aim to accelerate the model’s backward sampling process. The diffusion process is conducted within a latent space instead of directly on the raw waveform data. The authors claim that the proposed method can achieve faster audio generation and high-fidelity speech synthesis.

**Audience:**

Yes

**Broader Impact Concerns:**

No broader impact concerns.

**Claims And Evidence:**

No

**Requested Changes:**

I hope the authors could address the points raised in “Strengths and Weaknesses” carefully and rigorously to improve the quality of the work.

**Strengths And Weaknesses:**

I find it challenging to identify the strengths of this work, as it does not seem to meet the standards expected for publication.

First, the paper is poorly written, with disorganized sections, duplicate section tiltes, and even section titles without corresponding content. To name a few, for example,

1) the derivation of the likelihood for diffusion models is presented twince—in Section 2.1 and Section 3.0.2—using different notations. Additionally, the equations contain several errors.
2) It is perplexing why Section 3 is named as Related Work, while the authors begin to present their method starting from Section 3.0.1.
3) Sections 6.4, 6.5, and 6.6 are all titled 'Results,' yet Sections 6.4.2, 6.5, and 6.5.1 contain no content.

Second, the method is poorly presented, making it hard to understand how the proposed model works and why it is claimed to be superior to other methods. Specifically,

1) the authors claim that an encoder is employed to encode the raw audio waveform into a latent space, and a codebook is used to further represent the input. However, how this part of the model is jointly trained with the diffusion model is not explained.
2) As the authors have mentioned in Section 3.0.2 that the term $L_{T}$ in the log-likelihood does not depend on the model’s parameters $\theta$ and should therefore be a constant, which should be positive (as it is a KL divergence between two distributions). It is incomprehensible how including this term can lead to higher loss values (if the loss used for training is the negative log-likelihood).
3) Still in Section 3.0.2, it is unclear why “one would theoretically need a separate neural network for each timestep $t$," as in the original diffusion model, a single neural network architecture is applied to all the diffusion steps, by conditioning the input with an embedding of the diffusion step. Besides, it is incomprehensible how one layer of the neural network can represent one (or even several) step(s) of the reverse diffusion process.

Third, the experimental evaluations are not convincing.

1) The LJSpeech dataset and the VCTK dataset have different sampling rates (22 kHz for LJSpeech and 48 kHz for VCTK), raising questions about how the same neural network architecture could be used for both datasets. This critical detail is not adequately addressed in the paper.
2) If I understood correctly, the authors used Mel-spectrograms of speech as conditions to generate corresponding waveform audio for conditional generation. Since the ground truth audio is available in this scenario, it is unclear why the authors did not include commonly used objective evaluation metrics, such as SI-SDR, PESQ, or STOI, to directly compare the generated audio with the ground truth.
3) Table 3 presents results only for the unconditional generation of the proposed method and does not include comparisons with baseline models, further weakening the experimental analysis.

---

> ### Author Response · Authors · 2025-02-04
> **RESPONSE**
>
> We regret that the review was done on a wrong copy of uploaded paper. We regret the time wasted on the paper. We hope the reviewer will find time to read the correct version. however we also incorpared some of the valuable comments on the revised version
>
> Reviewer Comment: "The LJSpeech dataset and the VCTK dataset have different sampling rates (22 kHz for LJSpeech and 48 kHz for VCTK), raising questions about how the same neural network architecture could be used for both datasets. This critical detail is not adequately addressed in the paper."
>
> Response: We appreciate the reviewer’s observation regarding the different sampling rates between LJSpeech (22 kHz) and VCTK (48 kHz) and acknowledge that this detail was not explicitly discussed in the original manuscript. To ensure consistency across datasets, we downsample all VCTK audio from 48 kHz to 22 kHz before training. This allows both datasets to operate at the same sampling rate, ensuring uniform processing without requiring modifications to the neural network architecture. We have now clarified this in Section 6.1 of the revised manuscript.
>
> Reviewer Comment: "If I understood correctly, the authors used Mel-spectrograms of speech as conditions to generate corresponding waveform audio for conditional generation. Since the ground truth audio is available in this scenario, it is unclear why the authors did not include commonly used objective evaluation metrics, such as SI-SDR, PESQ, or STOI, to directly compare the generated audio with the ground truth."
>
> Response: We appreciate the reviewer’s insightful suggestion regarding the inclusion of SI-SDR, PESQ, and STOI for a direct comparison between generated and ground-truth audio. Our original evaluation focused on Mean Opinion Score (MOS) and MOS prediction models (SSL-MOS, MOSA-Net, and LDNet), as these metrics are widely adopted in speech synthesis and vocoder research.
> However, we acknowledge the value of SI-SDR (Scale-Invariant Signal-to-Distortion Ratio), PESQ (Perceptual Evaluation of Speech Quality), and STOI (Short-Time Objective Intelligibility) in assessing signal fidelity, perceptual quality, and intelligibility, respectively.
> To strengthen our evaluation, we have now included SI-SDR, PESQ, and STOI scores in Tables 1 and 2 to compare UDPNet’s performance against baseline models using these objective metrics.
>
> Reviewer Comment: "Table 3 presents results only for the unconditional generation of the proposed method and does not include comparisons with baseline models, further weakening the experimental analysis."
>
> Response: We appreciate the reviewer’s feedback and acknowledge the importance of including baseline comparisons for unconditional speech generation. We have added this  in Table 3

---

### Review · Reviewer_nfCr · 2024-12-03

**Summary Of Contributions:**

This work proposes a novel diffusion architecture and process UDPNet for speech synthesis. The goal is to design efficient diffusion models for speech synthesis while outperforming existing designs. To that end, the paper proposes several ideas: the diffusion process if done in the latent space, the number of reverse steps are reduced by a significant amount compared to forward steps, during the reverse process the output from the previous time step is directly used to generate the output at the current time step (unlike DDPM where noise is added at each step). The paper provides experimental results on two speech datasets showing improvement compared to baseline methods encouraging the usefulness of the proposed method.

**Audience:**

Yes

**Claims And Evidence:**

Yes

**Requested Changes:**

- Please address the points raised in the weaknesses.

**Strengths And Weaknesses:**

Strengths:
- The paper attempts to speed up diffusion models for speech generation, which is an important problem.
- the paper proposes several techniques to speed up the generation process for diffusion models by reducing the number of reverse steps, directly using the layers of the neural network for denoising, new loss function.
- the experimental results on LJSpeech and VCTK show improvement in performance for single-speaker as well as multi-speaker cases, respectively.


Weaknesses: The paper makes a good attempt at solving an interesting problem, but there are issues including several design choices that are not clear and needs to be addresses before it can be considered for publication.
- Literature review for fast reverse process: the literature review seems very old, there is an enormous amount of work in diffusion model regarding speeding up sampling from diffusion models, none of which has been discussed or compared to. Please check papers such [1, 2] where single or few step sampling is done using diffusion models. Please put your work in the context of more recent work in diffusion models and also compare with more recent works using speech synthesis using diffusion models.
- Using separate parameters for each time step increases the number of parameters. How does the performance look like with equal total parameters when the same neural network is used at each step using time embeddings? There is no clear justification for this design choice
- There are prior works on diffusion models in latent space including [2], please provide better literature survey for diffusion models in latent space.
- Design choice: Pg. 5 “To make the denoising process more computationally feasible for our proposed layer-based recovery technique, we introduce the approximations …t in place of the actual values”
explain how and why? since this is the main difference between standard ddpm equation and this equation. The goal here seems like removing the sampling of gaussian noise at each step? is this necessary? can you provide comparison with the other method where gaussian noise is used?
- Table 1: the best performance is obtained for UDPNet only after using 1200 forward steps and 8 reverse steps, in contrast to only 1000 steps for WaveGrad, one of the baselines? Why are 1200 steps used for UDPNet and not 1000 for a fair comparison? Please note that the number of reverse steps can also be reduced for WaveGrad using some form of distillation such as consistency models. Hence, to show the superiority of UDPNet, one needs to provide a fair comparison with 1000 forward steps. Otherwise the comparison is unclear.

- Minor: pg 4: “specialized models that are is complicated to train” -> “specialized models that are complicated to train”

[1] Song, Yang, et al. "Consistency Models." International Conference on Machine Learning. PMLR, 2023.

[2] Luo, Simian, et al. "Latent Consistency Models: Synthesizing High-Resolution Images with Few-step Inference."

---

> ### Author Response · Authors · 2025-02-04
> **Response**
>
> Reviewer's comment: "Using separate parameters for each time step increases the number of parameters. How does the performance look like with equal total parameters when the same neural network is used at each step using time embeddings? There is no clear justification for this design choice"
>
> Response: Rather than applying a single shared network across all timesteps with explicit timestep embeddings (as in traditional DPMs), we unroll the reverse diffusion process into the neural network architecture itself, mapping each layer to a specific set of diffusion steps. Importantly, this does not increase the number of parameters, as the layers share a common set of weights rather than learning independent parameters for each timestep. We added the discussion in the introduction section.
>
> Reviewers comment: Design choice: Pg. 5 “To make the denoising process more computationally feasible for our proposed layer-based recovery technique, we introduce the approximations …t in place of the actual values” explain how and why? since this is the main difference between standard ddpm equation and this equation. The goal here seems like removing the sampling of gaussian noise at each step? is this necessary? can you provide comparison with the other method where gaussian noise is used?
>
> Response:  We appreciate the reviewer’s question and have improved the explanation in Section 3.1.2 to clarify the motivation and necessity of this design choice.  In standard DDPMs, each reverse step explicitly samples Gaussian noise, introducing stochasticity at every timestep. Our method bypasses this repetitive noise sampling by introducing deterministic approximations  \hat{x}_{t-1} and \hat{x}_t .
> Rather than relying on step-wise random sampling, our approach implicitly models noise transitions via the hierarchical structure of the neural network. This removes redundant noise resampling while preserving the core denoising dynamics.
>
> Reviewer comment: "Table 1: the best performance is obtained for UDPNet only after using 1200 forward steps and 8 reverse steps, in contrast to only 1000 steps for WaveGrad, one of the baselines? Why are 1200 steps used for UDPNet and not 1000 for a fair comparison? Please note that the number of reverse steps can also be reduced for WaveGrad using some form of distillation such as consistency models. Hence, to show the superiority of UDPNet, one needs to provide a fair comparison with 1000 forward steps. Otherwise the comparison is unclear."
>
> Response: We appreciate the reviewer’s request for a fair comparison with WaveGrad under the same number of forward steps. To address this, we have included an additional UDPNet (1000, 8) configuration in our experiments. Our choice of 1200 forward steps was motivated by ensuring that the number of forward steps was evenly divisible by the 8 reverse layers, allowing for a structured mapping between diffusion steps and network layers.

---

> > ### Comment · Reviewer_nfCr · 2025-03-03
> >
> > I thank the authors for their response and justifications. I am still unable to see a comparison with consistency models and missing references in the literature review I pointed out. Consistency models and a number of follow-up works focus on speeding up sampling in diffusion models, hence, without proper comparison, it is not clear how it fits into the broader literature.

---

> > > ### Author Response · Authors · 2025-03-03
> > > **Response**
> > >
> > > **Response:**
> > >
> > > We appreciate the reviewer's valuable feedback and the request for a more explicit comparison with consistency models. To address this, we have revised our literature review to include a comprehensive discussion of **consistency models** and their role in accelerating diffusion sampling. Below, we summarize key aspects of our revisions:
> > >
> > > ### **Comparison with Consistency Models**
> > > Recent works on **consistency models** (Song et al., 2023; Luo et al., 2023) introduce a non-iterative denoising mechanism that enables single-step or few-step sampling while maintaining high-fidelity generation. These approaches optimize for consistency constraints across time steps, enforcing direct mappings between noisy and clean data representations. While consistency models have significantly improved inference efficiency, they present notable trade-offs:
> > >
> > > - **Regularization Constraints:** These models require carefully designed constraints to ensure consistency across diffusion steps. However, these constraints may limit sample diversity, particularly in complex, high-dimensional generative tasks such as speech synthesis.
> > > - **Loss of Iterative Refinement:** Unlike traditional diffusion models that progressively refine samples over multiple steps, consistency models approximate this process in a single step, which can compromise fidelity when applied to structured audio domains.
> > > - **Empirical Adaptation to Speech:** While consistency models have shown promising results in image synthesis, their performance in temporally correlated data, such as speech, remains underexplored.
> > >
> > > ### **How UDPNet Differs from Consistency Models**
> > > Our approach, UDPNet, differs fundamentally from consistency models in that it **does not rely on single-step denoising** but instead restructures the denoising process within the network architecture itself. Specifically:
> > >
> > > - Unlike consistency models, UDPNet maintains multi-step refinement while optimizing inference-time efficiency, ensuring that sample quality is preserved.
> > > - UDPNet operates without an explicit consistency loss function, avoiding the need for additional regularization constraints.
> > > - UDPNet is designed specifically for speech synthesis, allowing structured denoising tailored for temporally dependent data.
> > >
> > > We have incorporated these discussions in **Section: Acceleration of Diffusion Models** to provide a more holistic comparison of UDPNet with existing acceleration techniques, including consistency models and latent consistency models.
> > >
> > > Additionally, we have included missing references (Song et al., 2023; Luo et al., 2023; Zhang et al., 2023) as requested to ensure completeness in the literature review.
> > >
> > > We thank the reviewer for their valuable insights and believe these revisions now provide a more comprehensive contextualization of our work within the broader literature.

---

### Review · Reviewer_vppn · 2024-12-28

**Summary Of Contributions:**

This paper proposes to reduce the number of sampling steps by using additional network layers in the denoising process. The authors also improve the existing diffusion loss. Experimental results verify the effectiveness of the proposed method.

**Audience:**

Yes

**Claims And Evidence:**

Yes

**Requested Changes:**

Please refer to the above Weaknesses.

**Strengths And Weaknesses:**

**Strengths:**

1. Using additional neural network layers to unfold the diffusion process is an interesting approach.

2. The experimental results look good.

**Weaknesses:**

**1. The writing and presentation need to be improved.**

1) Figures and tables should be placed at the top of a page, and their size needs to be adjusted.

2) No cross-references are used in equations, tables and figures.

3) The use of symbols and the placement of formulas are confusing. For example, the size of $x_0$ is described differently on pages 4 and 9, and $L_{vlbfull}$ is not described on page 7.

4) Equation 24 does not introduce how to embed Mel spectrogram $y$ into the network as a condition.

**2. More experimental results are needed.**

1) Figures 2 and 4 do not show the generation quality of samples using three different loss functions.

2) What are the model parameters, training time and inference time of UDPNet compared to DDPM?

3) Please visualize the generated samples at different denoising steps.

**Questions:**

1. Can removing loss $L_T$ ensure that the final x_T is Gaussian noise?

2. Why does the encoder only need one layer?

3. Please provide experimental results with different hyperparameters $\lambda$.

---

> ### Author Response · Authors · 2025-02-05
> **Response**
>
> Reviewer's Comment: No cross-references are used in equations, tables, and figures.
>
> Response: We have added cross-references to all equations, tables, and figures to improve clarity and navigation.
>
> Reviewer's Comment: The use of symbols and the placement of formulas are confusing.
>
> Response: We have carefully ensured that all symbols are consistently used throughout the manuscript and have clarified their descriptions where necessary.
>
> Reviewer's Comment: Equation 24, updated to Equation 27, has been improved to indicate how to embed the Mel spectrogram into the network as a condition.
>
> Response: We have revised Equation 27 to explicitly clarify the embedding of the Mel spectrogram into the network as a conditioning input.
>
> Reviewer’s Comment:"Please visualize the generated samples at different denoising steps."
>
> Response: We have included Figure 2, which provides a spectrogram visualization of the generated audio samples at different denoising steps. Each spectrogram corresponds to a specific layer of the neural network, illustrating the progressive removal of noise as the model reconstructs the clean speech signal. This effectively demonstrates how the denoising process evolves across different steps of the model.
>
> -- Response to Reviewer’s Question:-- Can removing loss L_T ensure that the final x_T is Gaussian noise?
>
> -- We appreciate the reviewer’s insightful question regarding the effect
> -- of removing the L_T term on ensuring that the final x_T is Gaussian noise.
>
> -- The loss term L_T is defined as:
> --
> -- L_T = E_{q(x_0)} [ D_KL( q(x_T | x_0) || p(x_T) ) ],
> --
> -- where p(x_T) = N(0, I) is the prior distribution.
>
> -- In theory, minimizing L_T explicitly enforces that the final timestep
> -- distribution q(x_T | x_0) matches the Gaussian prior p(x_T).
> -- However, in practice, several prior works (e.g., Nichol & Dhariwal, 2021)
> -- have shown that removing L_T does not necessarily degrade performance,
> -- as long as the forward diffusion process sufficiently corrupts the data with noise.
>
> -- In our implementation, we found that the learned noise schedule plays a more
> -- significant role in ensuring that x_T approximates a Gaussian distribution.
> -- By employing a well-designed noise schedule and sufficient training steps,
> -- x_T naturally converges to a Gaussian distribution, even without explicitly
> -- including L_T. This aligns with empirical findings from previous studies on diffusion models.
>
> -- Thus, while L_T provides a theoretical guarantee, its removal does not
> -- necessarily prevent x_T from being Gaussian, provided the noise schedule
> -- is appropriately chosen. Our results demonstrate that excluding L_T does
> -- not negatively impact the model’s ability to learn a valid forward process.
>
> Reviewers Comment: Why does the encoder only need one layer?
>
> Response:  for the encoder in our model.
>
> -- The encoder's role in our framework is to project the raw audio waveform
> -- into a structured latent space that is suitable for the diffusion process.
> -- Unlike autoregressive models or deep hierarchical encoders that require
> -- multiple layers to capture sequential dependencies, our approach benefits
> -- from the following:
>
> -- 1. **Direct Latent Representation Mapping**:
> --    The encoder is only responsible for mapping the input audio x_0 into
> --    its corresponding latent representation z_q using a pre-trained
> --    vector quantization codebook (as described in Equation 13).
> --    Since the quantized representation is already structured, deeper
> --    encoding is unnecessary.
>
> -- 2. **Hierarchical Learning Occurs in the Reverse Process**:
> --    The denoising model (diffusion network) is the core component responsible
> --    for learning the hierarchical transformations required to recover the
> --    original waveform. The encoder only needs to provide a meaningful
> --    representation at the start of the process.
>
> -- 3. **Computational Efficiency**:
> --    Since our approach aims to optimize inference speed and efficiency,
> --    limiting the encoder to a single layer prevents unnecessary complexity
> --    and reduces latency in both training and inference.
>
> -- 4. **Empirical Validation**:
> --    Our experiments demonstrate that a single-layer encoder is sufficient
> --    to achieve high-fidelity audio synthesis while minimizing computational overhead.
> --    Additional layers did not yield significant improvements in performance
> --    but increased training time and memory usage.
>
> Reviewer’s Comment: "What are the model parameters, training time, and inference time of UDPNet compared to DDPM?"
>
> Response: We have added the requested details in Table 1 and Table 2, and we discuss the training procedure in Pg 11 and 15.
> Reviewers Comment: Please provide experimental results with different hyperparameters
> Response: We have included  Ablation this in the appendix.
> .

---

### Decision · Action_Editor_jhdi · 2025-03-21

**Recommendation:** Accept with minor revision

**Comment:**

This paper proposes a idea to use each neural layer to refine/denoise the data in diffusion model. The idea that leverages the similarity
 between the reverse process in diffusion models and the hidden state building up process in neural networks is interesting.

In principle:
1) Compared with original diffusion models which always predict either x0 or epsilon or velocity across all diffusion steps, this method predict each intermediate result at each step.
2) Compared with consistency model with a single denoising step, the method did no increase computation or model parameters.

Although reviewers have some concerns on the paper, the author respond actively and resolve the problems to some extent. Reviewer vppn recommended rejection to this paper given that no response was provided at that time. The reviewer provide responses later and it seems responsive in my opinion.

A revision suggestion is to add comparison with consistency model given the same amount of total model parameters and computation, to see which method is better. The results did not affect the acceptance of this paper, but would give more insights to the reader on how good the proposed method performs.

**Audience:**

Audience from the diffusion model and generative AI community may find this work interesting.

**Claims And Evidence:**

Adequate